# Pre-Training Multimodal Hallucination Detectors with Corrupted Grounding Data

**Spencer Whitehead**[1]*     **Jacob Phillips**[2]     **Sean Hendryx**[2]

[1]Microsoft     [2]Scale AI
spwhitehead@microsoft.com     {jacob.phillips, sean.hendryx}@scale.com

## Abstract

Multimodal language models can exhibit hallucinations in their outputs, which limits their reliability. The ability to automatically detect these errors is important for mitigating them, but has been less explored and existing efforts do not localize hallucinations, instead framing this as a classification task. In this work, we first pose multimodal hallucination detection as a sequence labeling task where models must localize hallucinated text spans and present a strong baseline model. Given the high cost of human annotations for this task, we propose an approach to improve the sample efficiency of these models by creating corrupted grounding data, which we use for pre-training. Leveraging phrase grounding data, we generate hallucinations to replace grounded spans and create hallucinated text. Experiments show that pre-training on this data improves sample efficiency when fine-tuning, and that the learning signal from the grounding data plays an important role in these improvements.

## 1   Introduction

The capabilities of Multimodal Language Models (MLMs) continue to increase [3, 25, 30], making it enticing to use them in a wide range of scenarios. However, questions around their reliability may limit this adoption [8, 29]. For instance, when serving as a multimodal assistant for users with visual impairments, incorrect answers to questions [40] or hallucinations in output descriptions [37] can have negative consequences as users may base decisions on these outputs.

A critical step towards mitigating hallucinations is accurately detecting them, and a well-trained hallucination detector can be employed in many different ways (*e.g.,* as a reward model for fine-tuning the MLM [42, 45] or as an output filter/re-ranker at inference time [10, 32]). In this work, we pose multimodal hallucination detection as a sequence labeling task where, given an image, prompt, and response, models must *localize* hallucinated spans in the response. In contrast to prior work (*e.g.,* [10]), we do not assume access to pre-defined spans to classify, which we argue is a more realistic setting as pre-defined spans are likely unavailable in real scenarios. We present a strong baseline detector for this task.

Further, training hallucination detectors requires fine-grained annotations, like error spans [10] or corrections [45], that can be non-trivial to collect and scale due to the need for human annotators and/or powerful teacher models. Hence, most effectively using this data is important. We benchmark the sample efficiency when fine-tuning on human annotations, showing much room for improvement.

Therefore, we propose a simple approach to increase the sample efficiency by pre-training on corrupted grounding data, which we automatically create. Using phrase grounding data [33, 48], we replace some grounded spans with hallucinated phrases from a text-only Language Model (LM). The

---

*Corresponding author. Work done while at Scale AI.

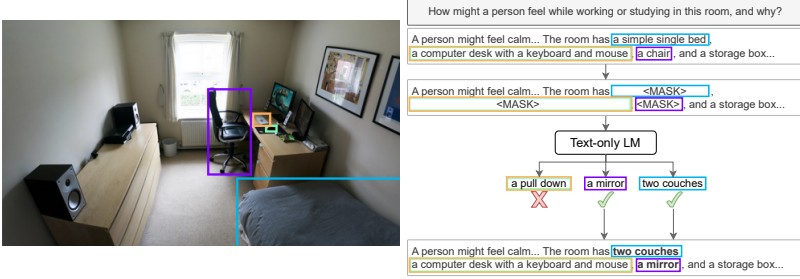

Figure 1: Our approach for creating corrupted grounding data to pre-train multimodal hallucination detectors. Examples of this data are in Appendix I.

LM does not take the image as input so it proposes phrases that are plausible for the text context but likely incorrect given the visual context. We find that pre-training on this corrupted grounding data improves sample efficiency when fine-tuning (*e.g.,* up to +7 F1 with 500 fine-tuning samples). We also show that using grounding annotations for our data is important, suggesting that grounding can offer a useful learning signal for training hallucination detectors.

In summary, our contributions are: 1) We formalize multimodal hallucination detection as a sequence labeling task and present a baseline. 2) We propose an approach to improve the sample efficiency of the detectors by creating corrupted grounding data and pre-training on this data. 3) Our experiments show that this improves sample efficiency when fine-tuning across different model and data scales. 4) We find that utilizing grounding data is important in our approach, suggesting that grounding offers a valuable learning signal for pre-training these detectors.

## 2   Hallucination Detection

**Task.** Given an image and associated prompt-response pair, the goal is to predict which text spans in the response are hallucinated and which are not. Prior work frames this as a *classification* task where pre-defined spans are given as input [10]. However, in an end-to-end setting, spans are either not provided or must be artificially imposed (*e.g.,* sentence boundaries). We explore hallucination *detection*, which we pose as a sequence labeling task where models predict a label for each token that indicates whether the token is part of a hallucinated segment. We adopt the binary setup from prior work [10], with non-hallucinated/hallucinated labels. We evaluate using span F1 scores for a given intersection-over-union (IoU) threshold, so models must identify span boundaries and classify the spans, much like other localization tasks [16, 22]. We compute macro F1 scores across the two classes to handle imbalances.

**Model.** We use a MLM as our base model and replace the next-token-prediction head with an output head that predicts a label based on the representation of each token from the base model. Since we predict per-token labels, we let transitions between labels in the token sequence demarcate the spans. This setup is compatible with a wide variety of base models. We use this modeling setup for both pre-training on our corrupted grounding data (Sec. 3) and fine-tuning (Sec. 4).

More details on the task and models are in Appendix F and Appendix H, respectively.

## 3   Corrupted Grounding Data

We want a scalable way to bolster the sample efficiency of hallucination detectors. Pre-training and transfer learning has been effective for improving downstream performance and sample efficiency in other areas (*e.g.,* [2]). However, pre-training requires more data and, as discussed, human annotations can be expensive to collect.

A promising alternative is to create synthetic or pseudo-labeled data that can be used for pre-training, which has been powerful for LMs [2, 9, 28]. In our setting, grounding data can be automatically created at large scales, albeit with some noise [12, 19, 44, 48]. Moreover, hallucinations and grounded phrases are linked since correctly grounded phrases are, by definition, not hallucinated. By replacing

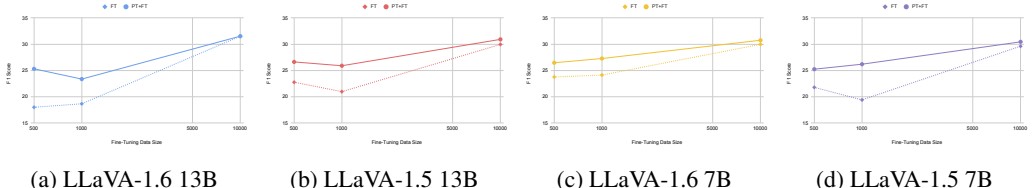

| (a) LLaVA-1.6 13B | (b) LLaVA-1.5 13B | (c) LLaVA-1.6 7B | (d) LLaVA-1.5 7B |

Figure 2: Sample efficiency of different models at 500, 1k, and 10k fine-tuning samples. Dotted lines are models that only fine-tune (FT), while solid lines are models that first pre-train on our data then fine-tune (PT+FT). Pre-training with our corrupted grounding data consistently improves the sample efficiency. Scores are listed in Appendix D.

grounded phrases with other phrases that are not aligned with the image, we can create text that contains hallucinations.

Shown in Fig. 1, we take multimodal data with grounding annotations and corrupt it to create hallucinated text. First, we mask out grounded spans and use a text-only LM to propose phrases to fill in the masked spans. This LM does not take the image as input, so it proposes phrases that are plausible for the *text context* but are likely incorrect for the *visual context*. We take measures to increase the likelihood that the proposals are hallucinations, such as restricting the LM from generating the original phrases and sampling during decoding to encourage more diversity [13]. Next, we randomly select a subset of the masked spans to fill in with the proposed phrases, keeping the original phrases for the remaining. We label any in-filled spans as hallucinated, while the remaining spans are labeled as non-hallucinated. Since most grounded spans tend to be noun phrases [33], the hallucinated labels may be sparse. Therefore, if a sentence contains any hallucinated spans, then we randomly decide whether to label the entire sentence as hallucinated. This noisy, corrupted data simulates hallucinations in the text that we can use to pre-train hallucination detectors. Approach details are in Appendix E and pre-training data analysis is in Appendix I.

## 4 Experiments

We experiment on M-HalDetect [10], a multimodal hallucination detection benchmark that has image-prompt-response triples with hallucinated span annotations (details in Appendix G). M-HalDetect has a training set of 11k samples and a test set of 3k. We fine-tune models on 500, 1k, and 10k subsets of the M-HalDetect training data to examine sample efficiency at distinct scales. We use the remaining 1k training samples as a validation set. We report F1 scores on the test set with an IoU threshold of 0.5 (Sec. 2).

For base models, we use LLaVA-1.5 and LLaVA-1.6 [24, 25], two strong and widely adopted MLMs. While structurally similar, they are distinct in important ways, such as their encoding of images, vision-language connector, and training data. For each model, we experiment with the 7B and 13B sizes to explore scaling. We do a light hyperparameter search and report the best result for each model at each data scale.

To create corrupted grounding data, we start from the Grounded Visual Chat dataset [48], which is automatically generated. We use 121k samples from this dataset. T5 [34] serves as our LM to propose hallucinated phrases since it is inexpensive to use and supports in-filling without prompt engineering.

Detailed settings are in Appendix E-H.

### 4.1 Benchmarking Detector Sample Efficiency

We explore sample efficiency on the detection task at different scales of fine-tuning data. We compare only fine-tuning (FT) to pre-training with our corrupted grounding data then fine-tuning (PT+FT). Qualitative examples are in Appendix I.

**FT baseline.** Looking at the FT results at 10k samples (*i.e.,* the full fine-tuning set), we see that all models achieve non-trivial F1 scores. The best performing detection model uses LLaVA-1.6 13B as

the base model, with 31.52% F1. These models serve as our strong baseline to which we compare our pre-training approach.

**Pre-training improves sample efficiency.** In Fig. 2, we see consistent improvements in sample efficiency across each of the models. For instance, with 500 samples, LLaVA-1.6 13B reaches 25.30% F1 with pre-training and 17.98% without. With this same model, the difference in performance between 500 and 10k samples decreases from 13.54% to 6.22% when pre-training. This suggests that by pre-training on our data, the model is able to make more effective use of the expensive human annotations. Similar observations hold for the other models as well. Finally, we see that our pre-training is most effective at lower scales (500, 1k), whereas the difference is less pronounced when fine-tuning on the full 10k samples. Though scaling up the pre-training data may improve this.

**Larger models tend to benefit more from pre-training at lower data scales.** Comparing Figs. 2a and 2b with Figs. 2c and 2d, at 500 samples, the difference between PT+FT and FT is larger for the 13B models. The 7B models also benefit from the pre-training (Figs. 2c and 2d), though the gap is less than the larger ones. This aligns with similar observations on pre-training reward models for LM alignment [2].

**Hallucination detection is a challenging task.** Based on Fig. 2, we see that when fine-tuning on the 10k training split, models have up to ∼33% F1 score. Although we do not know the upper bound for this detection task on M-HalDetect (*i.e.,* human performance), the combination of these scores and the qualitative examples we show in Appendix I.2 suggest that our models represent a strong baseline, but there is much room to improve the performance.

**Detection vs Classification.** Classification can be viewed as a subtask of detection. To demonstrate this, we adapt our fine-tuned detection models to perform classification on pre-defined spans by taking a majority vote over the predicted token labels in each given span. We present the results in Appendix B, where we find that our detection models can achieve 81.63% F1 on classification.

## 4.2 Ablations

**Grounded spans are important.** In Fig. 3, we evaluate masking out random spans instead of grounded ones to examine the need for grounding data. We see noticeably lower performance across each data scale. Interestingly, pre-training on this data even significantly lowers the performance when fine-tuning on 10k samples. This suggests that incorporating a notion of "*groundability*" into the pre-training data is important for improving sample efficiency when fine-tuning.

**Plausible hallucinations are necessary at smaller data scales.** We ablate our use of a LM to generate plausible hallucinated phrases by in-filling the grounded spans with random phrases. The curve in Fig. 3 illustrates that this also has a significant negative effect at smaller data scales, but is not as harmful as using random, ungrounded spans.

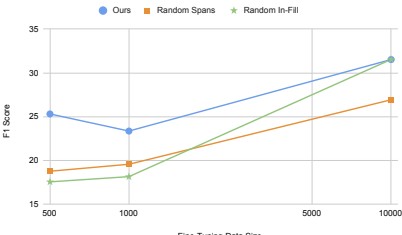

**Pre-training outperforms augmentation**. We also explore augmenting with our data rather than pre-training, with results in Appendix A. We find that pre-training outperforms augmentation likely, in part, due to differences in distribution and/or noise in our data.

We also explore freezing the base model during pre-training in Appendix A.

Figure 3: Ablations with LLaVA-1.6 13B for using grounding annotations and LMs for our data. Random Spans indicates that random text spans are masked and in-filled instead of grounded spans. Random In-Fill uses grounded spans but fills them in with random phrases.

## 5 Conclusions

Localizing hallucinations is important for mitigating them. We pose multimodal hallucination detection as a sequence labeling task and present a strong baseline detector. Given the cost of annotating hallucination detection data, we propose to improve the sample efficiency of detectors by creating corrupted grounding data and using this data for pre-training. We find that pre-training on this data improves sample efficiency across model and data scales, and that using grounded spans is important for these improvements.

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

# Appendix

**Table of Contents:**

## A  Further Ablations

We present ablations to further explore the design decisions of our approach.

**Augmentation.**  In our approach, we propose to pre-train with our data, but a straightforward alternative would be to instead augment the fine-tuning with our data. Tab. 1a shows that pre-training with our data benefits the model more than augmenting. In particular, the sample efficiency of augmentation is noticeably worse. Therefore, we pre-train the hallucination detectors to serve as a strong initialization on top of which we can fine-tune.

**Freezing weights.** Throughout our experiments, we initialize with a MLM backbone that has been trained on a wide array of multimodal data. We then fine-tune nearly the entire model (Sec. H) when adapting it to our task. Previous work has shown that fine-tuning can distort pre-trained features and degrade performance when transferring to different data distributions [15, 35]. Therefore, we also experiment with freezing the model backbone to preserve the rich features learned by the model and just tuning the output head during pre-training. The results of this are shown in Tab. 1b where we see that fully tuning the model is consistently more effective, suggesting that further adapting the model's learned features is useful.

## B  Classification Results

We argue that our detection task is more realistic than classification since pre-defined spans are likely unavailable in real settings. Further, the classification task could be viewed as a subtask of detection. We demonstrate this quantitatively by adapting our detection models to the classification task where we are given pre-defined spans to classify. To adapt our detectors to use pre-defined spans, we take a majority vote over the tokens in a given span to get its classification. We measure span-level, weighted F1 metric (wF1) to match [10].[2]

In Tab. 2, we examine the performance of our adapted FT models trained on our 10k train split versus the dedicated classification model from [10]. The base models differ between our adapted detection models and the classification model, so the results are not directly comparable. However, these results at least show the generality of the detection setup and that we can evaluate the classification performance of detection models as well.

We also show the effect of pre-training with our corrupted grounding data on the sample efficiency for classification in Fig. 4. Similar to detection, we observe improvements in this setting as well. Although, we expect the gap to be smaller for classification than when performing the more challenging detection task, which we do see in the plots.

---

[2]Our "span-level" is the same as "segment-level" from [10].

| Training | FT Data Scale | | |
| --- | --- | --- | --- |
| | 500 | 1k | 10k |
| FT-Aug | 11.75 | 13.25 | 19.07 |
| PT+FT | **25.30** | **23.35** | **31.52** |

(a) Comparison of augmenting the M-HalDetect data with our generated data (FT-Aug) vs pre-training on our data then fine-tuning on M-HalDetect (PT+FT). We present F1 scores across different M-HalDetect data scales.

| Learned in PT? | | FT Data Scale | | |
| --- | --- | --- | --- | --- |
| Base Model | Head | 500 | 1k | 10k |
| ✗ | ✓ | 13.08 | 14.48 | 22.70 |
| ✓ | ✓ | **25.30** | **23.35** | **31.52** |

(b) Effect of freezing the base model during our pre-training to preserve its learned features.

Table 1: Further ablations of our approach.

| Base Model | Params | Detection | wF1 |
| --- | --- | --- | --- |
| InstructBLIP | 7B | ✗ | 83.22 |
| LLaVA-1.5 | 7B | ✓ | 81.19 |
| LLaVA-1.6 | 7B | ✓ | 81.16 |
| LLaVA-1.5 | 13B | ✓ | 81.63 |
| LLaVA-1.6 | 13B | ✓ | 81.58 |

Table 2: Span-level weighted F1 scores (wF1) of the classification model from [10] (Detection ✗) versus our FT detection models adapted to use pre-defined spans (Detection ✓ ).

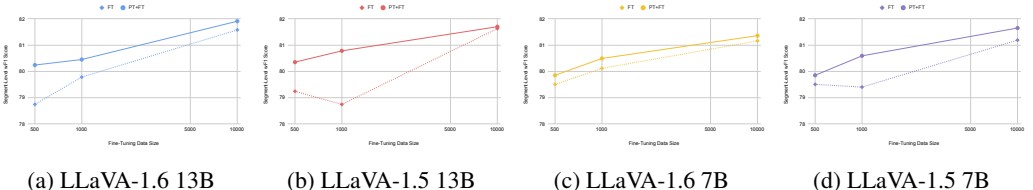

(a) LLaVA-1.6 13B     (b) LLaVA-1.5 13B     (c) LLaVA-1.6 7B     (d) LLaVA-1.5 7B

Figure 4: Classification sample efficiency of different models at 500, 1k, and 10k M-HalDetect fine-tuning samples. Dotted lines are models that only fine-tune (FT), while solid lines are models that first pre-train on our data then fine-tune (PT+FT).

## C  Prompting Proprietary LMs

We also attempted to explore prompting proprietary LMs (GPT-4 Turbo and GPT-4o) for our hallucination detection task. However, we had difficulties obtaining reliable token-level predictions from these models, much like observations on other sequence labeling tasks [39]. This may be an interesting direction for future work.

In lieu of the detection results, we present results for a simpler sentence classification task where the LM classifies whether each sentence contains a hallucination, which is akin to using pre-defined spans. We design a prompt for this task composed of instructions, an in-context example, and the target image-prompt-response triple as input. We use GPT-4 Turbo [29] and GPT-4o [30] as the LMs. For our hallucination detector, we run the detector to localized hallucinated spans. Then, if any span in a sentence is predicted as a hallucination, then we simply mark the sentence as containing a hallucination. We evaluate following the setup of the sentence classification task from [10].

The results in Tab. 3 show that the GPT models have strong performance for this sentence classification task and can slightly outperform the model from [10], which has been specifically fine-tuned for this task. Meanwhile, making a simple adaptation of our hallucination detector's outputs for this task yields performance beyond GPT-4 Turbo, but lower than that of GPT-4o. However, each of these other models do not localize hallucinations. As previously mentioned, using these LMs in our detection setting is challenging and warrants further exploration.

## D  Sample Efficiency Scores

Tab. 4 lists the scores for the plots in the main text for future comparisons.

| Model | Detection | wF1 |
|---|---|---|
| GPT-4 Turbo | ✗ | 73.16 |
| GPT-4o | ✗ | 79.57 |
| [10] | ✗ | 78.37 |
| Detector | ✓ | 74.60 |

Table 3: Sentence-level classification weighted F1 scores (wF1). We prompt GPT-4 Turbo and GPT-4o to obtain predictions. We also report the score from [10], which uses a model specifically fine-tuned for this task. "Detector" is our LLaVA-1.6 13B PT+FT detection model whose token-level outputs are used to get sentence-level predictions. Detection indicates whether a model is directly capable of localizing hallucinated spans.

| Model | FT Data Scale | | | Model | FT Data Scale | | | Model | FT Data Scale | | | Model | FT Data Scale | | |
|---|---|---|---|---|---|---|---|---|---|---|---|---|---|---|---|
| | 500 | 1k | 10k | | 500 | 1k | 10k | | 500 | 1k | 10k | | 500 | 1k | 10k |
| FT | 17.98 | 18.64 | **31.52** | FT | 22.75 | 20.96 | 29.95 | FT | 23.75 | 24.11 | 29.97 | FT | 21.78 | 19.38 | 29.61 |
| PT+FT | **25.30** | **23.35** | **31.52** | PT+FT | **26.62** | **25.89** | **30.91** | PT+FT | **26.46** | **27.27** | **30.75** | PT+FT | **25.23** | **26.18** | **30.44** |
| (a) LLaVA-1.6 13B | | | | (b) LLaVA-1.5 13B | | | | (c) LLaVA-1.6 7B | | | | (d) LLaVA-1.5 7B | | | |

Table 4: F1 scores for sample efficiency plots in Fig. 2.

# E    Corrupted Grounding Data

We start our data generation process from image-prompt-response triples with associated grounding annotations. Using these inputs, we create our corrupted grounding data by inserting hallucinations into the grounded spans. In this section, we detail the grounding data we use in our experiments, our settings for creating our corrupted grounding data, and present qualitative examples.

## E.1    Base Grounding Data

In general, our approach is compatible with phrase grounding datasets. We experiment with the Grounded Visual Chat (GVC) dataset [48] as our grounding data.[3] GVC is a large, open source grounded conversation dataset. This dataset has multimodal conversations in English and is open source under a CC BY NC 4.0 license for research purposes.[4] Each sample in this dataset includes an image from COCO [6, 22] and a conversation about the image. The conversations are from the LLaVA Visual Instruct 150k dataset [26], which are generated by GPT-4 and are comprised of multiple turns of prompt-response pairs. These conversations are then automatically annotated with visual grounding using GPT-4 as well. Since both the conversations and grounding annotations are automatically created, our approach operates on entirely synthetic data. We refer readers to [48] for more details.

For our experiments, we only utilize the first turn of the conversations in GVC. GVC contains 449,144 grounded spans over 121,909 samples for an average of 3.684 grounded spans per sample. We use 121,907 samples to create our data.

## E.2    Transforming to Corrupted Grounding Data

Sec. 3 discusses our corrupted grounding data generation approach. Here we provide more details for reproducibility.

We use T5 [34] as our LM for proposing hallucinations as it is easy to use and directly supports text in-filling. To balance quality and efficiency, we use T5-Base, which has 220M parameters and is licensed under an Apache-2.0 license. We access this model via HuggingFace [41].

Given a grounded response from a sample, we first randomly decide, with probability 0.95, whether or not to corrupt this sample. Since the grounded spans may be sparse in the text, this high probability helps to create more hallucinations while not removing all original correct samples.

---

[3]https://github.com/UX-Decoder/LLaVA-Grounding/releases/tag/train_data
[4]https://llava-vl.github.io/llava-grounding/

Next, for each grounded span in the sample, we mask out the span and input the masked sequence into the LM to fill in the masks. During decoding to fill the masks, we prevent the LM from generating the same phrases as the original grounded phrase by setting the probabilities of the original tokens (except stop words) to 0. Additionally, to encourage more diverse hallucination proposals, we perform multinomial sampling.

With the hallucination proposals from the LM, we randomly sample a subset of the proposals to replace the grounded phrases, while the rest of the masked segments are returned to their original phrases. We sample between 75% and 100% of the generated proposals as this subset. For example, if a response has 8 grounded spans, then we would sample 6-8 of them to replace with their hallucination proposals. We then transform these to our hallucination labels, where any grounded spans that have been replaced are labeled as hallucinated, the remaining are labeled as non-hallucinated. Since the responses may be long and grounded spans may be more sparse, if a sentence contains a hallucination, we randomly decide to label the entire sentence as hallucinated. We do this with probability 0.5.

For our random span and random in-fill ablations (Sec. 4.2), we largely maintain the exact same procedure as above. With random spans, given a response, we randomly sample sentences to insert hallucinations into, then randomly select a span of each sentence to mask and in-fill with the LM. With random in-fill, rather than using the LM, we sample between 1 and 5 words from a word frequency tool and use these as the hallucination proposals.[5]

## F    Hallucination Detection Task

For our task setup, models must localize hallucinated spans. Given annotations of which spans are hallucinated and which are not, we treat each contiguous span as one instance. To execute this task, models must predict their own spans and labels for each span. We compare the span boundaries and labels for evaluation.

We adopt an IoU-based metric to match spans between the ground truth and predictions with the same label. We use a minimum IoU threshold of 0.5 to consider two spans as matched. This guarantees unique matches between predictions and labels and establishes a sufficiently difficult task. We calculate per-class F1 scores and report macro F1 to handle class imbalance. This evaluation protocol is very similar to other localization tasks, such as object detetcion [22]. We do not use exact matches, like named entity recognition [16], to account for potential noise in the annotations.

## G    M-HalDetect Dataset Details

The M-HalDetect dataset [10] consists of image-prompt-response triples with span annotations on the responses. All language data is in English. We adopt the binary setting from [10], where we have non-hallucinated (labeled `Accurate`) and hallucinated (labeled `Inaccurate`). The images are sourced from the `val2014` split of COCO [6]. The prompts are curated by humans, while the responses are generated by InstructBLIP [7]. Responses are annotated by humans for hallucination span labels. We refer readers to [10] for more details.

We use the released version of the dataset, which has a train set of 10,979 samples and test set of 3,164 samples.[6] The annotations are released under a CC BY-NC 4.0 license and is for research purposes. We first split the train set into a 10,000 sample train split and 979 sample validation split. We also create 500 and 1,000 sample subsets of the 10k train split. The 500, 1k, and 10k splits are our different sizes of fine-tuning data for measuring sample efficiency.

## H    Detection Model Details

We adopt MLMs as our base models, which offer powerful multimodal backbones. We experiment with LLaVA-1.5 [24] and LLaVA-1.6 [25], and leverage the official implementation.[7] The implementation is under an Apache-2.0 license, while the checkpoints follow terms listed at the

---

[5]*"small"* set from https://github.com/rspeer/wordfreq/.

[6]https://github.com/hendryx-scale/mhal-detect

[7]https://github.com/haotian-liu/LLaVA

| Data Scale | LLaVA-1.6 13B | | LLaVA-1.5 13B | | LLaVA-1.6 7B | | LLaVA-1.5 7B | |
|---|---|---|---|---|---|---|---|---|
| | FT | PT+FT | FT | PT+FT | FT | PT+FT | FT | PT+FT |
| 500 | 2e-5, 12 | 2e-5, 12 | 2e-5, 12 | 2e-5, 12 | 2e-5, 12 | 2e-6, 12 | 2e-5, 12 | 2e-6, 12 |
| 1k | 2e-5, 12 | 2e-6, 12 | 2e-5, 12 | 2e-5, 12 | 2e-5, 12 | 2e-6, 12 | 2e-6, 12 | 2e-6, 12 |
| 10k | 8e-6, 6 | 2e-5, 6 | 8e-6, 6 | 8e-6, 6 | 2e-5, 3 | 8e-6, 6 | 8e-6, 6 | 8e-6, 6 |

Table 5: Learning rate and number of epochs for each model and data scale.

| Hyperparameter | Value |
|---|---|
| Batch Size | 128 |
| Optimizer | AdamW |
| Optimizer Momentum | (0.9, 0.999) |
| Weight decay | 0 |
| LR Scheduler | Cosine |
| LR Warmup Ratio | 0.03 |
| Context Length | 2048 |

Table 6: Model and Training Hyperparameters that stayed fixed throughout all experimentation runs.

official implementation. We use these resources for research purposes, in accordance with their licenses. We experiment with the 7B and 13B scales of each model and initialize our models from the instruction-tuned weights. To perform hallucination detection, we replace the next-token-prediction head of these models with an output head for our hallucination label space. Other architectural components remain the same.

We use a cross-entropy loss. We have also explored using a focal loss [21] for class imbalance in pre-training, but found this to perform worse. During both pre-training and fine-tuning, unless otherwise specified, the visual encoder is frozen while all other parameters are tuned.

We fix most of the hyperparameters throughout all our training runs (pre-training and fine-tuning), which we list in Tab. 6. We vary two hyperparamaters: learning rate and training epochs. For pre-training, we use a learning rate of 1e-6 and train for 3 epochs. For fine-tuning runs, we conduct a light hyperparameter search over combinations of learning rate, {2e-5, 8e-6, 2e-6}, and number of epochs, {3, 6, 12}. We choose these values based on early observations. For each data scale and model, we report the results from the best combination of hyperparameters. These best combinations are listed in Tab. 5 All models are trained on 8 NVIDIA A100 GPUs with DeepSpeed ZeRO-3.[8]

# I   Qualitative Analysis

## I.1   Corrupted Grounding Data

Fig. 5 shows the examples of our corrupted grounding data that we use for pre-training. In Fig. 5a, we see an example with a number of grounded spans (*e.g.,* "*a set of bottles*") that are masked an in-filled with hallucinations (*e.g.,* "*saucers*"). As a reminder, our algorithm for doing this randomly chooses a subset of the grounded spans to in-fill, so not all grounded spans are affected (*e.g.,* "*pizza on a baking tray*"). When creating hallucination labels from the corrupted response, for each span that is filled with a hallucinated phrase, we randomly decide whether to just label the span as hallucinated (Fig. 5b) or to label the entire sentence containing the span as hallucinated (Fig. 5a).

We observe some error cases in the corrupted grounding data. First, there are instances where the proposed hallucinated phrases are still somewhat valid for both the text context and image, such as "*excitement*" in Fig. 5c or "*what you see*" in Fig. 5d. Based on these observations, we have performed an analysis of 50 samples by examining the proposed hallucinated phrases within the text context along with the image and have discovered the following cases:

**Hallucination:** The proposed phrase fits in the text context and does not match the image (*e.g.,* Figs. 5a and 5b). This is our goal when proposing phrases and we find that hallucinatory phrases are 66% of those found in the corrupted grounding data.

---

[8]`https://github.com/microsoft/DeepSpeed`

**Semantic Match:** The proposed phrase semantically matches the image and still preserves original the meaning of the text (*e.g.,* "*excitement*" in Fig. 5c). These phrases are not true hallucinations, but can be marked as such, which introduces noise. We find 10% are semantic matches.

**Generic Phrase:** A less specific phrase is proposed so the text is less detailed, potentially making the text more ambiguous and less aligned with the image (*e.g.,* Fig. 5d)). Such phrases are about 18% of our proposed phrases.

**Other:** The proposed phrase is not a real word, makes the text incoherent, or other spurious errors. This noise makes up 6%.

Based on this analysis, the majority of the proposed phrases create actual hallucinations. However, there clearly is noise in our data, making it better-suited for pre-training. Some of this noise may be addressable via extra filtering, re-ranking candidates [11], or by generating hallucinations with more powerful MLMs [30]. However, our results show that there are still significant sample efficiency improvements despite such noise.

### I.2 Detection Output Examples

We present qualitative results in Figs. 6, 7, and 8. Each example is from LLaVA-1.6 13B fine-tuned on 500 samples from M-HalDetect. In Fig. 6 and Fig. 7, we instances where pre-training noticeably helps the model predict the correct spans. For both cases, the FT model has sparser span predictions, whereas the PT+FT model is able to predict more correct, contiguous spans. Fig. 8 shows a failure case where the PT+FT model only detects a small part of a hallucinated span whereas the FT model comes much closer to detecting the entire span, albeit somewhat sparsely.

## J  Related Work

Much focus has been placed on identifying, evaluating, and mitigating hallucinations in MLM outputs [4, 14, 17, 18, 20, 23, 31, 32, 37, 43, 45, 46, 47]. Concurrent with our work, [5] design a tool-based system to detect hallucinations in outputs across multiple multimodal tasks (*e.g.,* visual question answering [1], text-conditioned image generation [36]). This approach utilizes external tools to generate claims, which are labeled as hallucinated or not by an oracle MLM. The complexity and cost of running this pipeline and the tools involved (*e.g.,* GPT-4V [29], object detector [27], search engine) could make this difficult to use. We focus training models, without tools, to localize hallucinations in MLM outputs.

[10] release a hallucination detection benchmark with human annotations and propose a model for detecting hallucinations that treats this as a classification problem without localization. [38] generate synthetic hallucination data and use it to train an evaluator without localization. Here, we explore end-to-end detection, without pre-defined spans, and propose a method to improve the sample efficiency of the detectors with corrupted grounding data.

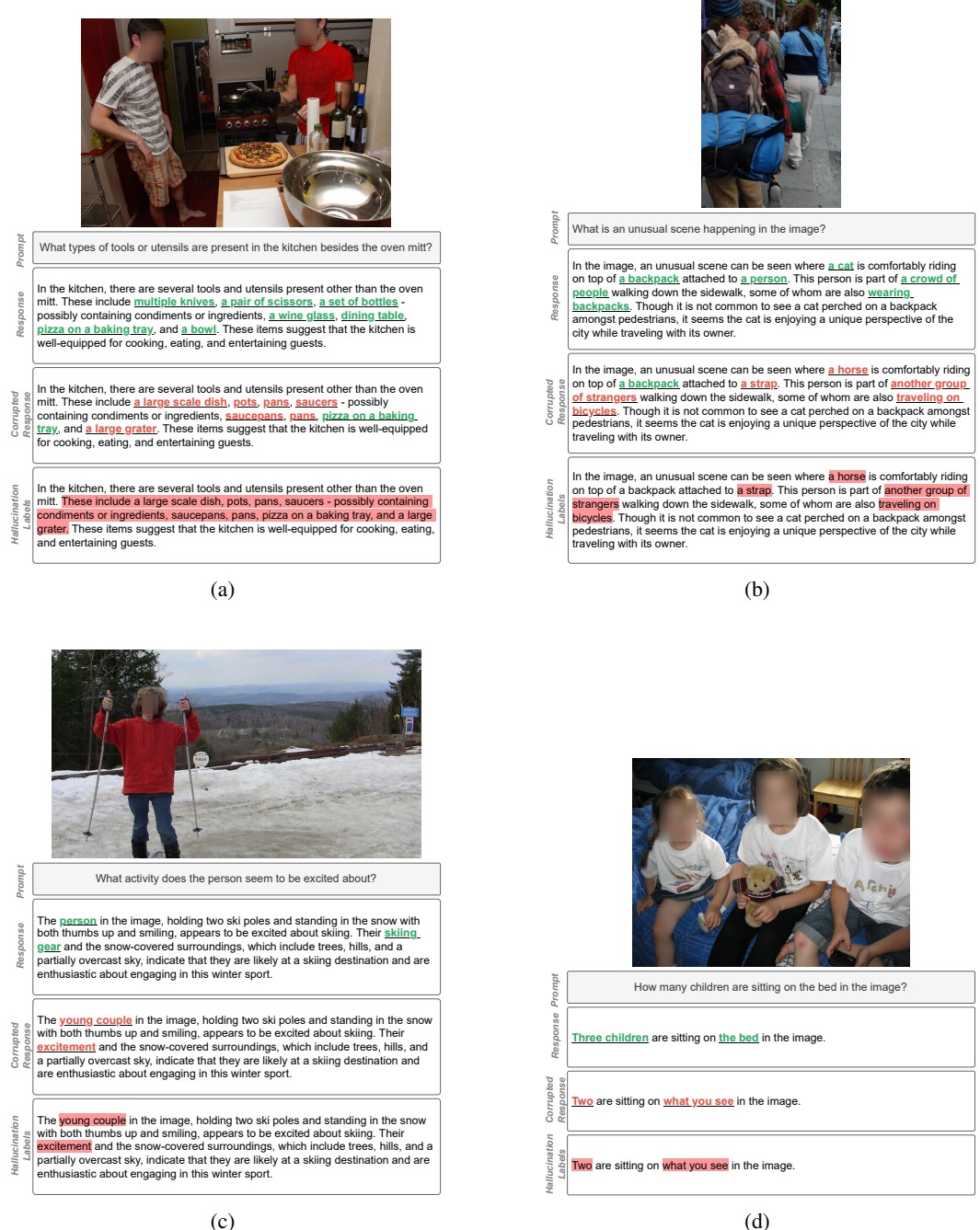

Figure 5: Examples of our corrupted grounding data. We show the prompt and original response with grounded spans (green), followed by our corrupted response with some hallucinations inserted for grounded spans (red), and then the final hallucination labels that we use for pre-training. For clarity, in the hallucination labels, we only highlight phrases marked as hallucinations.

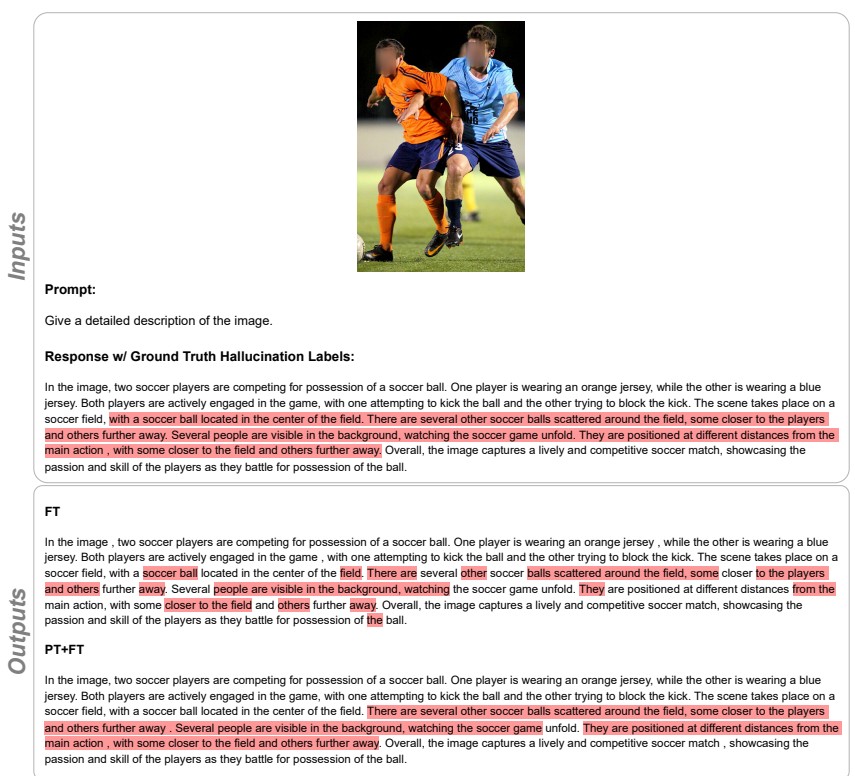

**Prompt:**

Give a detailed description of the image.

**Response w/ Ground Truth Hallucination Labels:**

In the image, two soccer players are competing for possession of a soccer ball. One player is wearing an orange jersey, while the other is wearing a blue jersey. Both players are actively engaged in the game, with one attempting to kick the ball and the other trying to block the kick. The scene takes place on a soccer field, with a soccer ball located in the center of the field. There are several other soccer balls scattered around the field, some closer to the players and others further away. Several people are visible in the background, watching the soccer game unfold. They are positioned at different distances from the main action , with some closer to the field and others further away. Overall, the image captures a lively and competitive soccer match, showcasing the passion and skill of the players as they battle for possession of the ball.

**FT**

In the image , two soccer players are competing for possession of a soccer ball. One player is wearing an orange jersey , while the other is wearing a blue jersey. Both players are actively engaged in the game , with one attempting to kick the ball and the other trying to block the kick. The scene takes place on a soccer field, with a soccer ball located in the center of the field. There are several other soccer balls scattered around the field, some closer to the players and others further away. Several people are visible in the background, watching the soccer game unfold. They are positioned at different distances from the main action, with some closer to the field and others further away. Overall, the image captures a lively and competitive soccer match, showcasing the passion and skill of the players as they battle for possession of the ball.

**PT+FT**

In the image, two soccer players are competing for possession of a soccer ball. One player is wearing an orange jersey, while the other is wearing a blue jersey. Both players are actively engaged in the game, with one attempting to kick the ball and the other trying to block the kick. The scene takes place on a soccer field, with a soccer ball located in the center of the field. There are several other soccer balls scattered around the field, some closer to the players and others further away . Several people are visible in the background, watching the soccer game unfold. They are positioned at different distances from the main action , with some closer to the field and others further away. Overall, the image captures a lively and competitive soccer match , showcasing the passion and skill of the players as they battle for possession of the ball.

Figure 6: Prediction examples from LLaVA-1.6 13B fine-tuned on 500 samples. We examine the outputs with (PT+FT) and without (FT) pre-training on our corrupted grounding data. For clarity, hallucinations are highlighed in red, while non-hallucinations are not highlighted.

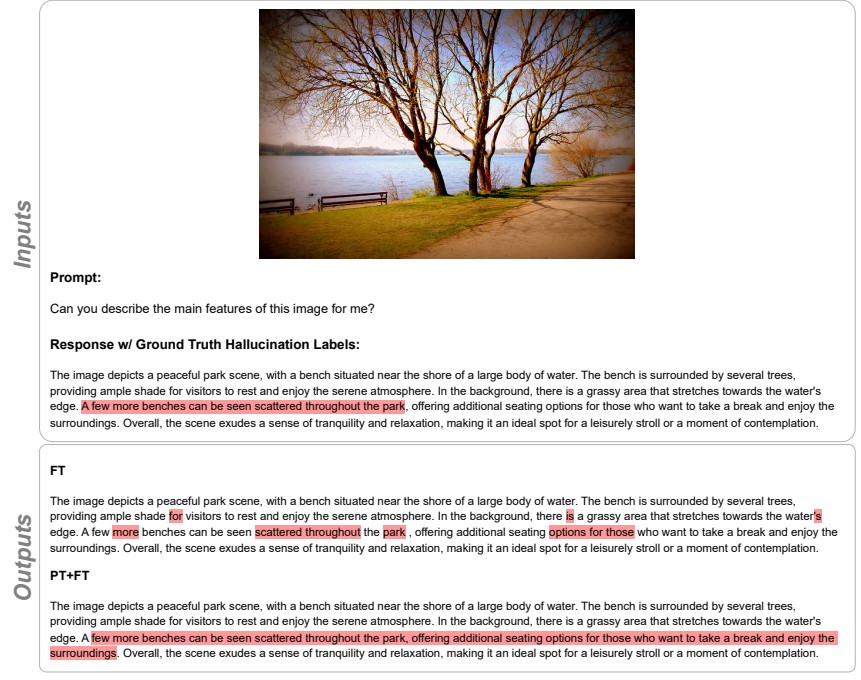

**Prompt:**

Can you describe the main features of this image for me?

**Response w/ Ground Truth Hallucination Labels:**

The image depicts a peaceful park scene, with a bench situated near the shore of a large body of water. The bench is surrounded by several trees, providing ample shade for visitors to rest and enjoy the serene atmosphere. In the background, there is a grassy area that stretches towards the water's edge. A few more benches can be seen scattered throughout the park, offering additional seating options for those who want to take a break and enjoy the surroundings. Overall, the scene exudes a sense of tranquility and relaxation, making it an ideal spot for a leisurely stroll or a moment of contemplation.

**FT**

The image depicts a peaceful park scene, with a bench situated near the shore of a large body of water. The bench is surrounded by several trees, providing ample shade for visitors to rest and enjoy the serene atmosphere. In the background, there is a grassy area that stretches towards the water's edge. A few more benches can be seen scattered throughout the park , offering additional seating options for those who want to take a break and enjoy the surroundings. Overall, the scene exudes a sense of tranquility and relaxation, making it an ideal spot for a leisurely stroll or a moment of contemplation.

**PT+FT**

The image depicts a peaceful park scene, with a bench situated near the shore of a large body of water. The bench is surrounded by several trees, providing ample shade for visitors to rest and enjoy the serene atmosphere. In the background, there is a grassy area that stretches towards the water's edge. A few more benches can be seen scattered throughout the park, offering additional seating options for those who want to take a break and enjoy the surroundings. Overall, the scene exudes a sense of tranquility and relaxation, making it an ideal spot for a leisurely stroll or a moment of contemplation.

Figure 7: Prediction examples from LLaVA-1.6 13B fine-tuned on 500 samples. We examine the outputs with (PT+FT) and without (FT) pre-training on our corrupted grounding data. For clarity, hallucinations are highlighed in red, while non-hallucinations are not highlighted.

Figure 8: Prediction examples from LLaVA-1.6 13B fine-tuned on 500 samples. We examine the outputs with (PT+FT) and without (FT) pre-training on our corrupted grounding data. For clarity, hallucinations are highlighed in red, while non-hallucinations are not highlighted.

