# OpenReview forum: "Pre-Training Multimodal Hallucination Detectors with Corrupted Grounding Data"
_NeurIPS.cc/2024/Workshop/SafeGenAi — SafeGenAi Poster_

### Official Review · Reviewer_wa4U · 2024-10-09
**Review for Pre-Training Multimodal Hallucination Detectors with Corrupted Grounding Data**

**Rating:** 6
**Confidence:** 4

**Review:**

Strength:
1. Novel Hallucination Localization:
One of the standout contributions of the paper is its focus on an underexplored area of hallucination detection—localizing hallucinated spans in multimodal models' responses. By framing hallucination detection as a sequence labeling task, the paper addresses a key gap in current research that mostly treats hallucination detection as a binary classification problem. This shift towards localization adds granularity and potential practical utility to hallucination detection efforts.
2. Scalable Pre-Training Approach:
The paper presents an effective pipeline for generating pre-training data by replacing grounded phrases with hallucinated ones. This approach is not only scalable but also a practical solution to the data scarcity problem in hallucination detection. The process of using corrupted grounding data allows for a significant improvement in sample efficiency during fine-tuning, making the proposed model more feasible for real-world applications where annotation costs are prohibitive.
3. Well-Organized Experiments:
The experimental design is clear and methodically demonstrates the viability of the proposed approach. The experiments are logically structured to assess the performance of the model in various conditions, such as different data scales and model sizes, which provides a comprehensive view of the model's robustness and efficiency. The inclusion of sample efficiency analyses at different fine-tuning data sizes is particularly useful for understanding the practical benefits of the method.

Weakness:
1. Noise in the Pre-Training Data:
While the scalability of the pre-training approach is a strength, the noise in the corrupted grounding data could have been better managed. Although the paper acknowledges the presence of noise and discusses its impact, even a simple noise reduction method, such as filtering through visual entailment models, could have been employed to minimize the noise, leading to cleaner pre-training data and potentially better results. (especially when the gains with 10K fine tuning data is quite small)
2. Analysis of the nature of hallucinated phrases in pre-training data:
Since most grounded phrases are noun-phrases, it concerns me that hallucinations pertaining to relationships, attributes, or actions(which MHalDetect's data includes) may be overlooked in the pre-training stage. A simple analysis or statistics on the nature of generated hallucinated phrases could be nice.
3. Low F1 scores:
A minor weakness( the arbitrary nature of span definitions in the M-HalDetect benchmark may contribute to this limitation, as the span boundaries lack fine-grained detail ).

---

### Official Review · Reviewer_5XfL · 2024-10-10
**The paper introduces a novel sequence labeling approach for detecting multimodal hallucinations and enhances pre-training with synthetic data, but lacks explanation in synthetic data generation choices, lacks diversity in baseline LVLMs and hallucination detection dataset..**

**Rating:** 5
**Confidence:** 5

**Review:**

### Strengths
- The authors introduce a new approach by framing multimodal hallucination detection as a sequence labeling task, which localizes hallucinated spans in text, unlike prior classification-based approaches​.
- By replacing grounded spans with hallucinated tokens generated from a text-only language model, the paper creates a synthetic dataset that enhances pre-training, leading to improved performance when fine-tuning​.

### Weaknesses
- Limited evaluation on M-Hal Detect, which only provides sub-sentence level hallucination labels. Does not cover the diversity of real-world hallucinations (object, attribute, relation).
- Why did the authors not include human evaluation of their synthetically generated corrupted grounding data?
- The reliance on corrupted grounding data for pre-training introduces a certain level of noise, as some hallucinated spans may still be plausible for the text context​. The experiments should also address some failure cases of their approach as they are relying on synthetic data for hallucination detection.
- The paper could have benefited from a more in-depth discussion on the trade-offs between pre-training with corrupted data and traditional method.
- How does the authors ensure the quality of the hallucinated phrases suggested by the T5 model? Reliance and validity of the corrupted grounding data should be addressed.
- Also, there are many state of the art LVLMs. The authors should have included performances of different types of LVLMs for their baseline models.

---

### Official Review · Reviewer_DkS9 · 2024-10-10
**Review for Multimodal Hallucination Detection Through Pre-Training on Corrupted Grounding Data**

**Rating:** 5
**Confidence:** 2

**Review:**

This study addresses the development of a pre-training technique aimed at enhancing hallucination detection tasks within multimodal language models. The problem of hallucination in MLMs is highly relevant, and the proposed method shows promise in mitigating these issues through a practical and scalable approach.

Pros:
1. Novelty: The framework of hallucination detection as a sequencial labeling is innovative and address the issue of localization more effectively than traditional classification methods.
2. Relevance: The problem of hallucinations in MLMs is timely and relevant as multimodal models are being increasingly adopted in real-world applications.
3. Practical approach: The use of corrupted grounding data for pre-training is a practical solution to the problem of acquiring expensive human annotations, showing significant improvements in sample efficiency.

Cons:
1. Resiability of synthetic data: The corrupted grounding data is automatically generated, which introduces noise and potential inaccuracies. While the results are promising, the reliance on synthetic data may limit the generalizability of the method to a variety of scenarios with more diverse and noisy inputs.

---

### Official Review · Reviewer_P9mK · 2024-10-10
**I think the paper researches an important topic, while more details should be included.**

**Rating:** 5
**Confidence:** 4

**Review:**

In this paper, the authors present an approach to formulate multimodal hallucination detection as a sequence labeling task. Also, they present a dataset for pre-training LLMs to save expensive human-labeled datasets. This paper is well-organized and easy to understand. The topic is also cutting-edge.

My concern is that the authors are suggested to provide more details and experiments in the main body of this paper. For example, the experiments compare the effectiveness of FT and PT + FT. However, a comparison with other baselines of hallucination detection should be provided to further validate the effectiveness of the proposals. Also, the strategy for generating hallucination data for pre-training is already very common and well-established, which may weaken the contribution of this paper.